# SnapBoost: A Heterogeneous Boosting Machine

**Thomas Parnell**[*]
IBM Research
Zürich, Switzerland
tpa@zurich.ibm.com

**Andreea Anghel**[*]
IBM Research
Zürich, Switzerland
aan@zurich.ibm.com

**Małgorzata Łazuka**
ETH Zürich
Zürich, Switzerland
lazukam@student.ethz.ch

**Nikolas Ioannou**
IBM Research
Zürich, Switzerland
nio@zurich.ibm.com

**Sebastian Kurella**
ETH Zürich
Zürich, Switzerland
kurellas@student.ethz.ch

**Peshal Agarwal**
ETH Zürich
Zürich, Switzerland
agarwalp@student.ethz.ch

**Nikolaos Papandreou**
IBM Research
Zürich, Switzerland
npo@zurich.ibm.com

**Haralampos Pozidis**
IBM Research
Zürich, Switzerland
hap@zurich.ibm.com

## Abstract

Modern gradient boosting software frameworks, such as XGBoost and LightGBM, implement Newton descent in a functional space. At each boosting iteration, their goal is to find the base hypothesis, selected from some base hypothesis class, that is closest to the Newton descent direction in a Euclidean sense. Typically, the base hypothesis class is fixed to be all binary decision trees up to a given depth. In this work, we study a Heterogeneous Newton Boosting Machine (HNBM) in which the base hypothesis class may vary across boosting iterations. Specifically, at each boosting iteration, the base hypothesis class is chosen, from a fixed set of subclasses, by sampling from a probability distribution. We derive a global linear convergence rate for the HNBM under certain assumptions, and show that it agrees with existing rates for Newton's method when the Newton direction can be perfectly fitted by the base hypothesis at each boosting iteration. We then describe a particular realization of a HNBM, SnapBoost, that, at each boosting iteration, randomly selects between either a decision tree of variable depth or a linear regressor with random Fourier features. We describe how SnapBoost is implemented, with a focus on the training complexity. Finally, we present experimental results, using OpenML and Kaggle datasets, that show that SnapBoost is able to achieve better generalization loss than competing boosting frameworks, without taking significantly longer to tune.

## 1 Introduction

Boosted ensembles of decision trees are the dominant machine learning (ML) technique today in application domains where tabular data is abundant (e.g., competitive data science, financial/retail industries). While these methods achieve best-in-class generalization, they also expose a large number of hyper-parameters. The fast training routines offered by modern boosting frameworks allow one to effectively tune these hyper-parameters and are an equally important factor in their success.

The idea of boosting, or building a strong learner from a sequence of weak learners, originated in the early 1990s [41], [21]. This discovery led to the widely-popular AdaBoost algorithm [22], which

---

[*]Equal contribution.

iteratively trains a sequence of weak learners, whereby the training examples for the next learner are weighted according to the success of the previously-constructed learners. An alternative theoretical interpretation of AdaBoost was presented in [23], which showed that the algorithm is equivalent to minimizing an exponential loss function using gradient descent in a functional space. Moreover, the same paper showed that this idea can be applied to arbitrary differentiable loss functions.

The modern explosion of boosting can be attributed primarily to the rise of two software frameworks: XGBoost [14] and LightGBM [27]. Both frameworks leverage the formulation of boosting as a functional gradient descent, to support a wide range of different loss functions, resulting in general-purpose ML solutions that can be applied to a wide range of problems. Furthermore, these frameworks place a high importance on training performance: employing a range of algorithmic optimizations to reduce complexity (e.g., splitting nodes using histogram summary statistics) as well as system-level optimizations to leverage both many-core CPUs and GPUs. One additional characteristic of these frameworks is that they use a second-order approximation of the loss function and perform an algorithm akin to Newton's method for optimization. While this difference with traditional gradient boosting is often glossed over, in practice it is found to significantly improve generalization [43].

From a theoretical perspective, boosting algorithms are not restricted to any particular class of weak learners. At each boosting iteration, a weak learner (from this point forward referred to as a *base hypothesis*) is chosen from some base hypothesis class. In both of the aforementioned frameworks, this class comprises all binary decision trees up to a fixed maximum depth. Moreover, both frameworks are *homogeneous*: the hypothesis class is fixed at each boosting iteration. Recently [18, 44, 17] have considered *heterogeneous* boosting, in which the hypothesis class may vary across boosting iterations. Promising results indicate that this approach may improve the generalization capability of the resulting ensembles, at the expense of significantly more complex training procedures.

The goal of our work is to build upon the ideas of [18, 44, 17], and develop a heterogeneous boosting framework with theoretical convergence guarantees, that can achieve better generalization than both XGBoost and LightGBM, without significantly sacrificing performance.

**Contributions.** The contributions of this work can be summarized as follows:

- We propose a Heterogeneous Newton Boosting Machine (HNBM), in which the base hypothesis class at each boosting iteration is selected at random, from a fixed set of subclasses, according to an arbitrary probability mass function $\Phi$.

- We derive a global linear convergence rate for the proposed HNBM for strongly convex loss functions with Lipschitz-continuous gradients. Our convergence rates agree with existing global rates in the special case when the base hypotheses are fully dense in the prediction space.

- We describe a particular realization of a HNBM, SnapBoost, that randomly selects between $K$ different subclasses at each boosting iteration: $(K - 1)$ of these subclasses correspond to binary decision trees (BDTs) of different maximum depths and one subclass corresponds to linear regressors with random Fourier features (LRFs). We provide details regarding how SnapBoost is implemented, with a focus on training complexity.

- We present experiments using OpenML [45] and Kaggle [7] datasets that demonstrate SnapBoost generalizes better than competing boosting frameworks, without compromising performance.

## 1.1 Related Work

**Heterogeneous Boosting.** In [44], the author proposes a heterogeneous boosting algorithm, KTBoost, that learns both a binary decision tree (BDT) and a kernel ridge regressor (KRR) at each boosting iteration, and selects the one that minimizes the training loss. It is argued that by combining tree and kernel regressors, such an ensemble is capable of approximating a wider range of functions than trees alone. While this approach shows promising experimental results, and was a major source of inspiration for our work, the complexity of the training procedure does not scale: one must learn multiple base hypotheses at every iteration. In [18], the authors derive margin-based generalization bound for heterogeneous ensembles. Specifically, the authors showed that the generalization error is bounded from above by the empirical margin error plus a weighted sum over the Rademacher complexities of the different base hypothesis subclasses. The weights correspond to how many times each subclass appears in the ensemble. This bound suggests that heterogeneous ensembles have desirable generalization properties, since by selecting a highly complex subclass fairly *infrequently*,

one may be able to reduce the empirical margin error without blowing up the complexity term. The same paper also proposes a boosting algorithm, DeepBoost, that chooses the base hypothesis at each boosting iteration by explicitly trying to minimize the aforementioned bound. The authors acknowledge that exploring the entire hypothesis space is computationally infeasible, and propose a greedy approach that is specific to decision trees of increasing depth.

**Randomized Boosting.** Stochastic behavior in boosting algorithms has a well-established history [24], and it is common practice today to learn each base hypothesis using a random subset of the features and/or training examples. Recently, a number of works have introduced additional stochasticity, in particular when selecting the base hypothesis class at each boosting iteration. In [36], a randomized gradient boosting machine was proposed that selects, at each boosting iteration, a subset of base hypotheses according to some uniform selection rule. The HNBM proposed in our paper can be viewed as a generalization of this approach to include (a) arbitrary non-uniform sampling of the hypothesis space and (b) second-order information. A form of non-uniform sampling of the hypothesis space was also considered in [17], however second-order information was absent.

**Ordered Boosting.** An orthogonal research direction tries to improve the generalization capability of boosting machines by changing the training algorithm to avoid *target leakage*. CatBoost [39] implements this idea, together with encoding of categorical variables using ordered target statistics, oblivious decision trees, as well as minimal variance example sampling [28].

**Deep Learning Approaches.** [38, 12, 32] have introduced differentiable architectures that are in some sense analogous to boosting machines. Rather than using functional gradient descent, these models are trained using end-to-end back-propagation and implemented in automatic differentiation frameworks, e.g., TensorFlow, PyTorch. While the experimental results are promising, a major concern with this approach is the comparative training and/or tuning time, typically absent from the papers. We compare the methods of our paper with one such approach in Appendix F and find that the deep learning-based approach is 1-2 orders of magnitude slower in terms of tuning time.

## 2 Heterogeneous Newton Boosting

In this section we introduce heterogeneous Newton boosting and derive theoretical guarantees on its convergence under certain assumptions.

### 2.1 Preliminaries

We are given a matrix of feature vectors $X \in \mathbb{R}^{n \times d}$ and a vector of training labels $y \in \mathcal{Y}^n$, where $n$ is the number of training examples and $d$ is the number of features. The $i$-th training example is denoted $x_i^T \in \mathbb{R}^d$. We consider an optimization problem of the form:

$$\min_{f \in \mathcal{F}} \sum_{i=1}^{n} l(y_i, f(x_i)), \tag{1}$$

where loss function $l : \mathcal{Y} \times \mathbb{R} \to \mathbb{R}^+$, and $\mathcal{F}$ is a particular class of functions to be defined in the next section. We assume that the loss function $l(y, f)$ is twice differentiable with respect to $f$, $l'(y, f)$ and $l''(y, f)$ denote the first and second derivative respectively, and satisfies the following assumptions:

**Assumption 1** ($\mu$-strongly convex loss). *There exists a constant $\mu > 0$ such that $\forall y, f_1, f_2$:*

$$l(y, f_1) \geq l(y, f_2) + l'(y, f_2)(f_1 - f_2) + \frac{\mu}{2}(f_1 - f_2)^2 \iff l''(y, f) \geq \mu$$

**Assumption 2** ($S$-Lipschitz gradients). *There exists a constant $S > 0$ such that $\forall y, f_1, f_2$:*

$$|l'(y, f_1) - l'(y, f_2)| \leq S|f_1 - f_2| \iff l''(y, f) \leq S$$

Examples of loss functions that satisfy the above criteria are the standard least-squares loss: $l(y, f) = \frac{1}{2}(y - f)^2$, as well as L2-regularized logistic loss: $l(y, f) = \log(1 + \exp(-yf)) + \frac{\lambda}{2}f^2$ for $\lambda > 0$.

### 2.2 Heterogeneous Newton Boosting

We consider a heterogeneous boosting machine in which the base hypothesis at each boosting iteration can be drawn from one of $K$ distinct subclasses. Let $\mathcal{H}^{(k)}$ denote the $k$-th subclass for $k \in [K]$ which satisfies the following assumption:

---

**Algorithm 1** Heterogeneous Newton Boosting Machine

---

1: **initialize:** $f^0(x) = 0$
2: **for** $m = 1, \ldots, M$ **do**
3:     Compute vectors $[g_i]_{i=1,\ldots,n}$ and $[h_i]_{i=1,\ldots,n}$
4:     Sample subclass index $u_m \in \{1, 2, \ldots, K\}$ according to probability mass function $\Phi$
5:     Fit base hypothesis: $b_m = \arg\min_{b \in \mathcal{H}^{(u_m)}} \sum_{i=1}^n h_i \left(-g_i/h_i - b(x_i)\right)^2$
6:     Update model: $f^m(x) = f^{m-1}(x) + \epsilon b_m(x)$
7: **end for**
8: **output:** $f^M(x)$

---

**Assumption 3** (Subclass structure). *For $k \in [K]$*

$$\mathcal{H}^{(k)} = \left\{ \sigma b(x) : b(x) \in \bar{\mathcal{H}}^{(k)}, \sigma \in \mathbb{R} \right\},$$

*where $\bar{\mathcal{H}}^{(k)}$ is a finite class of functions $b : \mathbb{R}^d \to \mathbb{R}$ that satisfy $\sum_{i=1}^n b(x_i)^2 = 1$.*

We note that while the subclasses used in practice (e.g., trees) may well be infinite beyond a simple scaling factor, in practice they are finite when represented in floating point arithmetic. We now consider the optimization problem (1) over the domain:

$$\mathcal{F} = \left\{ \sum_{m=1}^M \alpha_m b_m(x) : \alpha_m \in \mathbb{R}, b_m \in \left\{ \mathcal{H}^{(1)} \cup \mathcal{H}^{(2)} \ldots \cup \mathcal{H}^{(K)} \right\} \right\}. \tag{2}$$

Our proposed method for solving this optimization problem is presented in full in Algorithm 1. At each boosting iteration, we randomly sample one of the $K$ subclasses according to a given probability mass function (PMF) $\Phi$. The probability that the $k$-th subclass is selected is denoted $\phi_k$. Let $u_m \in [K]$ denote the sampled subclass index at boosting iteration $m$. The base hypothesis to insert at the $m$-th boosting iteration is determined as follows:

$$b_m = \arg\min_{b \in \mathcal{H}^{(u_m)}} \left[ \sum_{i=1}^n l(y_i, f^{m-1}(x_i) + b(x_i)) \right] \approx \arg\min_{b \in \mathcal{H}^{(u_m)}} \left[ \sum_{i=1}^n h_i \left(-g_i/h_i - b(x_i)\right)^2 \right], \tag{3}$$

where the approximation is obtained by taking the second-order Taylor expansion of $l(y_i, f^{m-1}(x_i) + b(x_i))$ around $f^{m-1}(x_i)$ and the expansion coefficients are given by $g_i = l'(y_i, f^{m-1}(x_i))$ and $h_i = l''(y_i, f^{m-1}(x_i))$. In practice, an L2-regularization penalty, specific to the structure of the subclass, may also be applied to (3). It should be noted that (3) corresponds to a standard sample-weighted least-squares minimization which, depending on the choice of subclasses, enables one to reuse a plethora of existing learning algorithms and implementations [2]. Intuitively, the algorithm chooses the hypothesis from the randomly selected subclass $\mathcal{H}^{(u_m)}$ that is closest (in a Euclidean sense) to the Newton descent direction, and dimensions with larger curvature are weighted accordingly. To ensure global convergence, the model is updated by applying a learning rate $\epsilon > 0$:

$$f^m(x) = f^{m-1}(x) + \epsilon b_m(x).$$

In practice, $\epsilon$ is normally treated as a hyper-parameter and tuned using cross-validation, although some theoretical insight on how it should be set to ensure convergence is provided later in the section.

### 2.3 Reformulation as Coordinate Descent

While boosting machines are typically implemented as formulated above, they are somewhat easier to analyze theoretically when viewed instead as a coordinate descent in a very high dimensional space [36, 17]. Let $\bar{\mathcal{H}} = \bar{\mathcal{H}}^{(1)} \cup \bar{\mathcal{H}}^{(2)} \cup \ldots \cup \bar{\mathcal{H}}^{(K)}$ denote the union of the finite, normalized subclasses defined in Assumption 3. Furthermore, let $b_j \in \bar{\mathcal{H}}$ denote an enumeration of the hypotheses for $j \in [|\bar{\mathcal{H}}|]$ and $I(k) = \{j : b_j \in \bar{\mathcal{H}}^{(k)}\}$ denote the set of all indices corresponding to normalized

hypotheses belonging to the $k$-th subclass. Let $B \in \mathbb{R}^{n \times |\bar{\mathcal{H}}|}$ be a matrix with entries given by $B_{i,j} = b_j(x_i)$. Then we can reformulate our optimization problem (1) over domain (2) as follows:

$$\min_{\beta \in \mathbb{R}^{|\bar{\mathcal{H}}|}} L(\beta) \equiv \min_{\beta \in \mathbb{R}^{|\bar{\mathcal{H}}|}} \sum_{i=1}^{n} l(y_i, B_i \beta),$$

where $B_i$ denotes the $i$-th row of $B$. In this reformulation, the model at iteration $m$ is given by:

$$\beta^m = \beta^{m-1} + \epsilon \sigma_{j_m}^* e_{j_m}, \tag{4}$$

where $e_j$ denotes a vector with value 1 in the $j$-th coordinate and 0 otherwise, $\sigma_j$ is the descent magnitude, and $j_m$ is the descent coordinate. For a given $j$, the magnitude $\sigma_j^*$ is given by:

$$\sigma_j^* = \min_{\sigma} \sum_{i=1}^{n} h_i \left(-g_i/h_i - \sigma B_{i,j}\right)^2 = -\nabla_j^2 L(\beta^{m-1})^{-1} \nabla_j L(\beta^{m-1}), \tag{5}$$

and the descent coordinate is given by:

$$j_m = \arg\min_{j \in I(u_m)} \left[ \sum_{i=1}^{n} h_i \left(-g_i/h_i - \sigma_j^* B_{i,j}\right)^2 \right] = \arg\max_{j \in I(u_m)} \left[ \left| \nabla_j^2 L(\beta^{m-1})^{-1/2} \nabla_j L(\beta^{m-1}) \right| \right]. \tag{6}$$

Further details regarding this reformulation are provided in Appendix G.

## 2.4 Theoretical Guarantees

In order to establish theoretical guarantees for Algorithm 1, we adapt the theoretical framework developed in [36] to our setting with non-uniform sampling of the subclasses as well as the second-order information. In particular, we derive a convergence rate that depends on the following quantity:

**Definition 1** (Minimum cosine angle). The minimum cosine angle $0 \leq \Theta \leq 1$ is given by:

$$\Theta = \min_{c \in Range(B)} \left\| \left[ \cos(B_{\cdot j}, c) \right]_{j=1 \ldots, |\bar{\mathcal{H}}|} \right\|_{\Phi}, \tag{7}$$

where $B_{\cdot j}$ denotes the $j$-th column of the matrix $B$ and $\|x\|_{\Phi} = \sum_{k=1}^{K} \phi_k \max_{j \in I(k)} |x_j|$.

The minimum cosine angle measures the expected *density* of base hypotheses in the prediction space. A value close to 1 indicates that the Newton direction can be closely fitted to one of the base hypotheses, and a value close to 0 the opposite.

In order to prove global convergence of Algorithm 1, we will need the following technical lemma:

**Lemma 1.** *Let $\mathbb{E}_m[.]$ denote expectation over the subclass selection at the $m$-th boosting iteration and let $\Gamma(\beta) = \left[ \nabla_j^2 L(\beta)^{-1/2} \nabla_j L(\beta) \right]_{j=1 \ldots, |\bar{\mathcal{H}}|}$ then the following inequality holds:*

$$\mathbb{E}_m \left[ \Gamma_{j_m}(\beta^{m-1})^2 \right] \geq \left\| \Gamma(\beta^{m-1}) \right\|_{\Phi}^2. \tag{8}$$

The proof is provided in Appendix H. With this result in hand, one can prove the following global linear convergence rate for Algorithm 1:

**Theorem 2.** *Given learning rate $\epsilon = \mu/S$ then:*

$$\mathbb{E} \left[ L(\beta^M) - L(\beta^*) \right] \leq \left( 1 - \frac{\mu^2}{S^2} \Theta^2 \right)^M \left( L(\beta^0) - L(\beta^*) \right), \tag{9}$$

*where the expectation is taken over the subclass selection at all boosting iterations.*

*Proof.* For clarity, we provide only a sketch of the proof, with the full proof in Appendix I. Starting from the coordinate update rule (4) we have:

$$L(\beta^m) = L \left( \beta^{m-1} - \epsilon \sigma_{j_m}^* e_{j_m} \right) \leq L(\beta^{m-1}) - \frac{\mu}{2S} \Gamma_{j_m}(\beta^{m-1})^2, \tag{10}$$

where the inequality is obtained by applying the mean value theorem and using Assumptions 1 and 2. Now taking expectation over the $m$-th boosting iteration and applying Lemma 1 we have:

$$\mathbb{E}_m\left[L(\beta^m)\right] \leq L(\beta^{m-1}) - \frac{\mu}{2S}\left\|\Gamma(\beta^{m-1})\right\|_\Phi^2 \leq L(\beta^{m-1}) - \frac{\mu}{2S^2}\left\|\nabla L(\beta^{m-1})\right\|_\Phi^2, \qquad (11)$$

where the second inequality is due to Assumptions 2 and 3. We then leverage Assumption 1 together with Proposition 4.4 and Proposition 4.5 from [36] to obtain the following lower bound:

$$\left\|\nabla L(\beta^{m-1})\right\|_\Phi^2 \geq 2\mu\Theta^2\left(L(\beta^{m-1}) - L(\beta^*)\right) \qquad (12)$$

Then, by subtracting $L(\beta^*)$ from both sides of (11), plugging in (12), and following a telescopic argument, the desired result is obtained. $\qquad\square$

We note that in the case where $\Theta = 1$ (i.e., implying there always exists a base hypothesis that perfectly fits the Newton descent vector) the rate above is equivalent to that derived in [31] for Newton's method under the same assumptions.

## 3  SnapBoost: A Heterogeneous Newton Boosting Machine

In this section, we describe SnapBoost, a realization of a HNBM that admits a low-complexity implementation. SnapBoost is implemented in C++ and uses OpenMP for parallelization and Eigen [5] for linear algebra. The algorithm is exposed to the user via a sklearn-compatible Python API.

### 3.1  Base Hypothesis Subclasses

At each boosting iteration, SnapBoost chooses the subclass of base hypotheses to comprise binary decision trees (BDTs) with probability $p_t$ or linear regressors with random Fourier features (LRFs) with probability $(1 - p_t)$. Furthermore, if BDTs are selected, the maximum depth of the trees in the subclass is chosen uniformly at random between $D_{min}$ and $D_{max}$, resulting in $K = N_D + 1$ unique choices for the subclass at each iteration, where $N_D = D_{max} - D_{min} + 1$. The corresponding PMF is given by: $\Phi = [\frac{p_t}{N_D}, \ldots, \frac{p_t}{N_D}, 1 - p_t]$. Note that the PMF $\Phi$ is fully parameterized by $p_t$, $D_{min}$ and $D_{max}$. A full list of hyper-parameters is provided in Appendix B.

### 3.2  Binary Decision Trees

In a regression tree, each node represents a test on a feature, each branch the outcome of the test and each leaf node a continuous value. The tree is trained using all or a subsample of the examples and features in the train set, where the example/feature sampling ratios ($r_n$ and $r_d$) are hyper-parameters. In order to identify the best split at each node, one must identify the feature and feature value which, if split by, will optimize (3). The tree-building implementation in SnapBoost is defined in three steps as follows. Steps 1 and 2 are performed only *once* for all boosting iterations, whereas Step 3 is performed on each node, for each boosting iteration at which a BDT is chosen.

**Step 1.** We sort the train set for each feature [26, 37, 42]. This step reduces the complexity of finding the best split at each node, which is a critical training performance bottleneck. However, it also introduces a one-off overhead: the sort time, which has a complexity of $O(dn\log(n))$.

**Step 2.** We build a compressed representation of the input dataset to further reduce the complexity of finding the best split at each node. We use the sorted dataset from Step 1 to build a histogram [14, 46] for each feature. Since we store the histogram bin index using a single byte, the number of histogram bins $h$ can be at most 256, and thus typically $h \ll n$. For each feature, its histogram bin edges are constructed before boosting begins, by iterating over the feature values and following a greedy strategy to balance the number of examples per bin. The complexity of building this histogram is $O(dn)$. Each histogram bin also includes statistics necessary to accelerate the computation of the optimal splits. While the bin edges remain fixed across boosting iterations, these statistics are continually recomputed during tree-building.

**Step 3.** The actual construction of the tree is performed using a depth-first-search algorithm [16]. For each node, two steps are performed: a) finding the best split, and b) initializing the node children. The complexity of step a) is $O(dr_dh)$: instead of iterating through the feature values of each example,

we iterate over the histogram bin edges. In step b), we first assign the node examples to the children, an operation of complexity $O(n_{node})$, where $n_{node}$ is the number of examples in the node being split. Then, we update the bin statistics, a step of complexity $O(n_{node}dr_d)$. Assuming a complete tree of depth $D$, the overall complexity of step 3 is $O(2^D dr_d h + dr_d n r_n D)$ for each boosting iteration at which a BDT is chosen.

### 3.3 Linear Regressors with Random Fourier Features

We use the method proposed in [40] to learn a linear regressor on the feature space induced by a random projection matrix, designed to approximate a given kernel function. The process is two-fold:

**Step 1.** First, we map each example $x \in \mathbb{R}^d$ in the train set $X$ to a low-dimensional Euclidean inner product space using a randomized feature map, $z$, that uniformly approximates the Gaussian radial basis kernel $\mathcal{K}(x, x') = \exp(-\gamma||x - x'||^2)$. The feature map $z : \mathbb{R}^d \to \mathbb{R}^c$ is defined as $z(x) = \sqrt{2/c}[\cos(\xi_1^T x + \tau_1), ... \cos(\xi_c^T x + \tau_c)]^T$, where the weights $\xi_i$ are i.i.d samples from the Fourier transform of the Gaussian kernel and the offsets $\tau_i$ uniformly drawn from $[0, 2\pi]$. The complexity of projecting the train set onto the new feature space is essentially given by the multiplication of the feature matrix $X \in \mathbb{R}^{n \times d}$ with the weights matrix $\xi \in \mathbb{R}^{d \times c}$, an operation of complexity $O(ndc)$. Similarly to the tree histograms, the randomized weights and offsets are generated only *once*, for all boosting iterations. The dimensionality of the projected space, $c$, is a hyper-parameter and is typically chosen as $c < 100$.

**Step 2.** Using the projection of the train set as input $X' \in \mathbb{R}^{n \times c}$, we solve the sample-weighted least-squares problem defined in (3), adding L2-regularization as follows: $\sum_{i=1}^{n} h_i \left( y_i' - w^T x_i' \right)^2 + \alpha \|w\|^2$, where $y_i' = -g_i/h_i$ are the regression targets. Given the low dimensionality, $c$, of the new feature space, we solve the least-squares problem by computing its closed-form solution $(X'^T X' + \alpha I)^{-1} X' y'$. The complexity of computing this solution is dominated by the complexity of the $X'^T X'$ operation $O(nc^2)$ or the complexity of the inversion operation $O(c^3)$. As $c \ll n$, the complexity of this step is $O(nc^2)$, for each boosting iteration at which a LRF is chosen.

Which step dominates the overall complexity of SnapBoost strongly depends on the range of tree depths, $D_{min}$ and $D_{max}$, the dimensionality of the projected space, $c$, as well as the PMF, $\Phi$, that controls the mixture. When performance is paramount, one may enforce constraints on $\Phi$ (e.g., $p_t \geq 0.9$) to explicitly control the complexity by favoring BDTs over LRFs or vice-versa.

## 4 Experimental Results

In this section, we evaluate the performance of SnapBoost against widely-used boosting frameworks.

**Hardware and Software.** The results in this section were obtained using a multi-socket server with two 20-core Intel(R) Xeon(R) Gold 6230 CPUs @2.10GHz, 256 GiB RAM, running Ubuntu 18.04. We used XGBoost v1.1.0, LightGBM v2.3.1, CatBoost v.0.23.2 and KTBoost v0.1.13.

**Hyper-parameter Tuning.** All boosting frameworks are tuned using the successive halving (SH) method [30]. Our SH implementation is massively parallel and leverages process-level parallelism, as well as multi-threading within the training routines themselves. Details regarding the SH implementation and the hyper-parameter ranges can be found in Appendix C and Appendix D respectively.

### 4.1 OpenML Benchmark

We compare XGBoost, LightGBM and SnapBoost across 10 binary classification datasets sourced from the OpenML platform [45]. Details regarding the characteristics of the datasets as well as their corresponding preprocessing steps are presented in Appendix A. Since the datasets are relatively small (between 10k and 20k examples), 3x3 nested stratified cross-validation was used to perform hyper-parameter tuning and to obtain a reliable estimate of the generalization loss. For each of the 3 outer folds, we perform tuning using cross-validated SH over the 3 inner folds. Some of the datasets also exhibit class imbalance, thus a sample-weighted logistic loss is used as the training, validation and test metric. The test losses (averaged over the 3 folds) are presented in Table 1. We observe that XGBoost does not win on any of the 10 datasets (average rank 2.6), LightGBM wins on 2 (average rank 2.2), whereas SnapBoost wins on 8/10 of the datasets (average rank 1.2).

Table 1: Average test loss using 3x3 nested cross-validation for the OpenML datasets.

| ID | Name | Examples | Features | XGBoost | LightGBM | SnapBoost |
|----|------|----------|----------|---------|----------|-----------|
| 4154 | CreditCardSubset | 14240 | 30 | 3.9990e-01 | 4.3415e-01 | **3.8732e-01** |
| 1471 | eeg-eye-state | 14980 | 14 | 1.3482e-01 | 1.4184e-01 | **1.2966e-01** |
| 4534 | PhishingWebsites | 11055 | 30 | 7.3332e-02 | **7.1488e-02** | 7.2481e-02 |
| 310 | mammography | 11183 | 6 | 2.6594e-01 | 2.7083e-01 | **2.6437e-01** |
| 734 | ailerons | 13750 | 40 | 2.5902e-01 | **2.5667e-01** | 2.5808e-01 |
| 722 | pol | 15000 | 48 | 3.6318e-02 | 3.5973e-02 | **3.4507e-02** |
| 1046 | mozilla4 | 15545 | 5 | 1.6973e-01 | 1.6623e-01 | **1.6300e-01** |
| 1019 | pendigits | 10992 | 16 | 1.8151e-02 | 1.9637e-02 | **1.8057e-02** |
| 959 | nursery | 12960 | 8 | 2.3469e-04 | 2.3107e-07 | **7.0620e-08** |
| 977 | letter | 20000 | 16 | 3.0621e-02 | 2.9400e-02 | **2.6005e-02** |
| | | | Average Rank: | 2.6 | 2.2 | **1.2** |

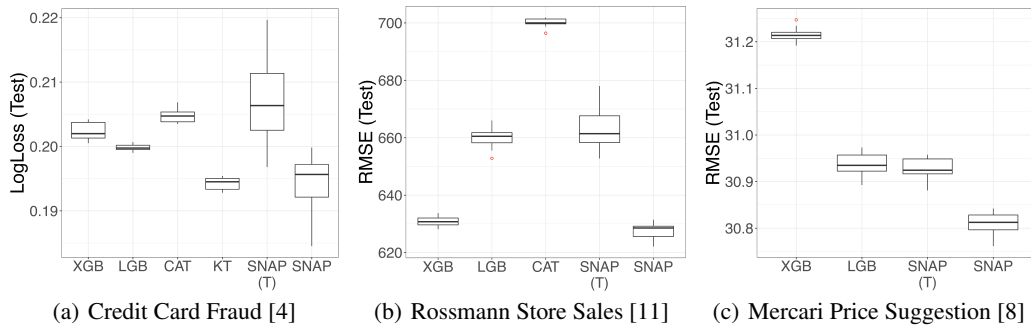

(a) Credit Card Fraud [4]  (b) Rossmann Store Sales [11]  (c) Mercari Price Suggestion [8]

Figure 1: Test loss for the different boosting frameworks (10 repetitions with different random seeds).

**Statistical Significance.** When comparing a number of ML algorithms across a large collection of datasets, rather than applying parametric statistical tests (such as Student's t-test) on a per-dataset basis, it is preferable to perform non-parametric tests across the collection [19]. Firstly, we apply the Iman and Davenport's correction of the Friedman omnibus test [29] to verify differences exist within the family of 3 algorithms ($p < 0.002$). Secondly, we perform pairwise testing using the Wilcoxon signed-rank test [13] (correcting for multiple hypotheses via Li's procedure [33]) to verify differences exist between the algorithms themselves. We find that the null hypothesis can be safely rejected when comparing SnapBoost with XGBoost ($p < 0.004$) and LightGBM ($p < 0.02$). However, when comparing XGBoost and LightGBM, the null hypothesis cannot be rejected ($p > 0.36$).

## 4.2 Kaggle Benchmark

In order to evaluate the generalization capability and performance of SnapBoost on more realistic use-cases we use 3 datasets from the Kaggle platform. Details of the datasets, as well as the preprocessing pipeline that was used are provided in Appendix A. Since these datasets are relatively large, we perform a single train/validation/test split. Hyper-parameter tuning (via SH) is performed using the train and validation sets. Once the tuned set of hyper-parameters is obtained, we re-train using the combined train and validation set and evaluate on the test set. The re-training is repeated 10 times using different random seeds, in order to quantify the role of stochastic effects on the test loss.

In Figure 1, we compare XGBoost (XGB), LightGBM (LGB), CatBoost (CAT), KTBoost (KT) and SnapBoost in terms of test loss [3]. To quantify the effect of including LRFs in the ensemble, we present results for SnapBoost restricted only to BDTs (SNAP (T)) as well as the unrestricted version that uses both BDTs and LRFs (SNAP). We observe that in all 3 datasets, SnapBoost generalizes better than the frameworks that use only BDTs (XGBoost, LightGBM and CatBoost). In Figure 1(a), we observe that SnapBoost achieves a similar test loss to KTBoost, that uses BDTs and KRRs.

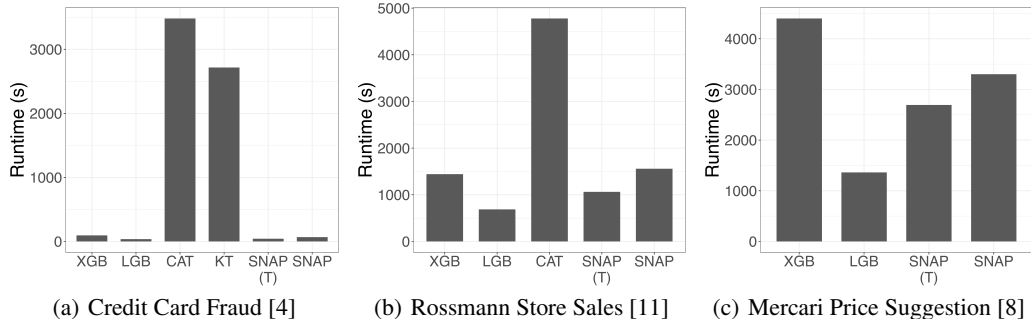

| (a) Credit Card Fraud [4] | (b) Rossmann Store Sales [11] | (c) Mercari Price Suggestion [8] |

Figure 2: End-to-end experiment time (tuning, re-training and evaluation) for all frameworks.

Table 2: Tuned SnapBoost hyper-parameter values.

| Dataset | $p_t$ | $D_{min}$ | $D_{max}$ |
|---|---|---|---|
| Credit Card Fraud | 0.912 | 1 | 1 |
| Rossmann Store Sales | 0.925 | 18 | 19 |
| Mercari Price Suggestion | 0.934 | 17 | 17 |

Next, in Figure 2, we compare the frameworks in terms of experimental time. We observe that, while the SnapBoost time is comparable to that of XGBoost and LightGBM, both CatBoost and KTBoost are significantly slower. This behaviour is expected for KTBoost since it (a) learns two base hypotheses at every iteration and (b) is implemented using scikit-learn components. For CatBoost, ordered boosting is known to introduce overheads when using a small number of examples[4]. Thus, it suffers significantly when tuned using the successive halving algorithm, which by construction evaluates a large number of configurations with very few examples.

Finally, in Table 2 we provide the tuned values of $p_t$, $D_{\min}$ and $D_{\max}$ for SnapBoost for the three Kaggle datasets. A full list of tuned hyper-parameter values for all frameworks is provided in Appendix E. We see that in all cases $D_{\min}$ and $D_{\max}$ are either very close or identical, indicating the improvement in generalization is *not* coming from using BDTs of varying depth. This is in agreement with Figure 1, in which we saw that SnapBoost restricted to BDTs (SNAP (T)) does not necessarily beat competing frameworks. In contrast, we see that the probability of selecting a LRF at each iteration $(1 - p_t)$ is between 7% and 9%. Thus, we conclude that the improved generalization capability of SnapBoost (SNAP) can be attributed to its use of LRFs alongside decision trees.

## 5   Conclusion

In this paper, we have presented a Heterogeneous Newton Boosting Machine (HNBM), with theoretical convergence guarantees, that selects the base hypothesis subclass stochastically at each boosting iteration. Furthermore, we have described an implementation of HNBM, SnapBoost, that learns heterogeneous ensembles of BDTs and LRFs. Experimental results on 13 datasets indicate that SnapBoost provides state-of-the-art generalization, without sacrificing performance. As a next step, we plan to explore different choices for both the probability mass function, $\Phi$, as well the base hypothesis subclasses. Additionally, we plan to further enhance the performance of SnapBoost by taking advantage of GPUs.

## Broader Impact

Boosting machines are generally considered most effective in application domains involving large amounts of tabular data. We have directly encountered use-cases in the retail, financial and insurance industries, and there are likely to be many others. Such examples include: credit scoring, residential mortgage appraisal, fraud detection and client risk profiling.

The tables used to train these models may contain sensitive personal information such as gender, ethnicity, health, religion or financial status. It is therefore critical that the algorithms used do not *leak* such information. In particular, an adversary should not be able to exploit a trained model to discover sensitive information about an individual or group. While we do not address these concerns in this paper, efforts are ongoing in the research community to develop privacy-preserving boosting machines [34].

Given the application domains where boosting machines are currently deployed, another important issue is fairness. Formally, we would like that certain statistics regarding the decisions produced by the trained model are consistent across individuals or groups of individuals. This definition imposes new constraints, which our training algorithms must be modified to satisfy. While this problem has received a significant amount of attention from the community in general, only a few works have looked at designing boosting machines that satisfy fairness constraints [20, 25]. Given the widespread use of boosting machines in production systems, this is a topic worthy of future investigation.

## Footnotes

[2] The supplemental material contains exemplary code for Algorithm 1 that uses generic scikit-learn regressors.

[3]Experiments that took longer than 8 hours were killed and do not appear in the plots.

[4]https://github.com/catboost/catboost/issues/505

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
