[Supplementary Material]

# A  Datasets

**OpenML.** The OpenML datasets are identified by their unique ID and can be downloaded programmatically using the OpenML API (`openml.datasets.get_dataset(ID)`). The IDs of the 10 datasets used in this work, as well as the number of examples and features, are provided in Table 1 in the main manuscript. Categorical features are encoded using scikit-learn's label encoder (`sklearn.preprocessing.LabelEncoder`). All of the datasets correspond to binary classification problems, with varying degrees of class imbalance. Stratified sampling (`sklearn.cross_validation.StratifiedKFold`) is used to construct the outer and inner folds for the nested cross-validation. The loss function used for training and evaluation is sample-weighted logistic loss (`sklearn.metrics.log_loss`).

**Rossmann Store Sales.** We download the raw data programmatically using the Kaggle API, which produces two files: `train.csv` and `store.csv`. Both files are read into pandas data frames and the missing values are replaced with zeros. We then follow several preprocessing steps inspired by an existing Kaggle kernel[5].

Firstly, we merge the train data frame with the store data frame, on the `Store` column. The resulting data frame is then sorted in ascending order by date. We then filter the data to exclude any stores that are not open, or have 0 sales. Next, we perform label encoding of the three categorical variables `StoreType`, `Assortment` and `StateHoliday`. We then extract four numeric features (month, year, day, and week of year) from the date feature. We create a feature corresponding to the number of months since the competition was open, and a similar feature corresponding to how many months a promotion has been running. We create one additional binary feature indicating whether the month is in the promotion interval. We then extract the `Sales` column as the labels and apply a logarithmic transformation. After all the pre-processing steps described above, the data matrix has 20 features.

While the prediction is always performed in the logarithmic domain, when evaluating the models we transform both the labels and the model predictions back into their original domain. The loss function used for training and evaluation is the standard root mean-squared error (`sklearn.metrics.mean_squared_error`). To create the train/validation/test split, we first extract all rows corresponding to the month of July. The extracted rows are then split 50/50 to form the validation and test set using the `train_test_split` function from scikit-learn with seed 42. The remaining rows are used for training only. The number of examples used for training, validation and test are 758762, 42788, and 42788, respectively.

**Mercari Price Suggestion.** We download the raw data programmatically using the Kaggle API, which produces the file `train.tsv`. We then follow several preprocessing steps inspired by an existing Kaggle kernel[6].

Firstly, we remove the products with price 0. Next, we replace missing values in the `name`, `category_name` and `item_description` columns with a constant string. We then *clean* these 3 columns, by 1) removing non-alpha characters, 2) converting to lower-case, and 3) applying scikit-learn's `CountVectorizer` with English stop-words and the maximum number of features set to 30. Next, we perform target encoding on the `brand_name` feature (`data['brand_name'].map(data.groupby('brand_name')['price'].mean()))` and we encode the `shipping` column using one-hot encoding (`pandas.get_dummies`). After all the pre-processing steps described above, the data matrix has 98 features.

Then, we extract the `price` column as the labels and apply a logarithmic transformation. The root mean squared error loss function is used for training and evaluation, with labels and predictions transformed back to the original domain. We then run L1-normalization on the rows and perform an 80/20 trainval/test split (with seed 42). The trainval set is then split 70/30 to generate the train and validation sets (with seed 42). The number of examples used for training, validation and test are 829729, 355599, and 296333 respectively.

**Credit Card Fraud.** We download the raw data programatically using the Kaggle API, which produces the file `creditcard.csv`. We extract the 31-st column as the binary labels and remove the first column `Time`. With the remaining columns we apply scikit-learn's `StandardScaler` followed

by L1-normalization of the rows. After all the pre-processing steps described above, the data matrix has 31 features.

We then perform a stratified 75/25 trainval/test split (seed 42), followed by a 70/30 train/val split (seed 42). The number of examples used for training, validation and test are 149523, 64082, and 71202 respectively. Since the data is highly imbalanced, for training and evaluation we use the sample-weighted logistic loss. The sample weights are computed using the `compute_sample_weight` function from scikit-learn (`sklearn.utils.class_weight`), using the `balanced` option.

## B   Hyper-Parameters of SnapBoost

In the following we list the hyper-parameters of the SnapBoost algorithm. We highlight in **bold** the hyper-parameters that typically require tuning when performing hyper-parameter optimization.

- ***num_round*** (int): the number of boosting iterations.
- *objective* ('mse', 'logloss'): the loss function optimized by the boosting algorithm.
- ***learning_rate*** (float): the learning rate of the boosting algorithm.
- *random_state* (int): the random seed used at training time.
- ***colsample*** (float): the fraction of features to be subsampled at each boosting iteration.
- ***subsample*** (float): the fraction of examples to be subsampled at each boosting iteration.
- ***lambda_l2*** (float): L2-regularization parameter applied to the tree leaf values.
- *early_stopping_rounds* (int): the number of boosting iterations used by early stopping.
- *base_score* (float): the initial prediction of all examples.

- ***tree_probability*** (float): the probability of selecting a tree at a boosting iteration.
- ***min_max_depth*** (int) : the minimum max_depth of a tree in the ensemble.
- ***max_max_depth*** (int): the maximum max_depth of a tree in the ensemble.
- *use_histograms* (bool): whether the tree uses histogram statistics or not.
- *hist_nbins* (int): number of histogram bins if *use_histograms* is `True`.
- *tree_n_threads* (int): the number of threads used to train the trees.

- ***alpha*** (float): the regularizer of the ridge regressor.
- ***fit_intercept*** (bool): whether to fit the intercept of the ridge regressor or not.
- *ridge_n_threads* (int): the number of threads used to train the ridge regressor.

- ***gamma*** (float): the gamma value of the Gaussian radial basis kernel.
- ***n_components*** (**c**) (int): the dimension of the randomized feature space.
- *kernel_n_threads* (int): the number of threads used to compute the dataset projection onto the new randomized feature space.

## C   Hyper-Parameter Optimization Method: Successive Halving

To perform hyper-parameter tuning we use the successive halving (SH) method from [30]. SH begins by training and evaluating a large number of hyper-parameter configurations using only a small fraction of the training examples (otherwise referred to as *resource*). The configurations are then ranked according to their validation loss and only the best-performing configurations are carried forward into the next *stage*, in which they are trained using a larger resource. This process repeats until the final stage, where all remaining configurations are trained using the maximal resource (i.e., the full train set). The general idea is that bad configurations can be eliminated in the earlier stages, without consuming a significant number of CPU cycles. Our implementation of SH is massively parallel and

**Algorithm 2** Successive Halving [30] with process-level and thread-level parallelism.

---
1: **Input:** initial number of configurations $n_0$
2: **Input:** elimination rate $\eta$
3: **Input:** minimum resource $r_{min}$
4: **Input:** number of processor cores `num_cores`
5: Determine number of stages $s_{max} = \lfloor -\log_\eta(r_{min}) \rfloor$
6: **Assert:** $n_0 \geq \eta^{s_{max}}$
7: Initialize set $C$ by sampling $n_0$ configurations at random
8: **for** $i = 0, 1, \ldots, s_{max}$ **do**
9:     Set number of configurations in this stage: $n_i = \lfloor n_0\eta^{-i} \rfloor$
10:     Set resource in this stage: $r_i = \eta^{i-s_{max}}$
11:     Set number of processes in this stage: $p_i = \min(\text{num\_cores}, |C|)$
12:     Set number of threads in this stage: $t_i = \lfloor \text{num\_cores}/p_i \rfloor$
13:     Populate input queue $Q_{in}$ with all configurations $c \in C$
14:     **parfor** $p = 0, 1, \ldots, p_i$ **do**
15:         Start new process with $t_i$ threads
16:         **while** $Q_{in}$ is not empty **do**
17:             Pull configuration $c$ from $Q_{in}$
18:             Train using fraction $r_i$ of training examples and compute validation loss $l$
19:             Push $(c, l)$ pair into output queue $Q_{out}$
20:         **end while**
21:     **end parfor**
22:     Sort output queue $Q_{out}$ by validation loss
23:     Update set $C$ to comprise the $n_i/\eta$ configurations with lowest validation loss
24: **end for**
25: **Output:** Configuration in $C$ with lowest validation loss

---

leverages both process-level parallelism (across configurations) and thread-level parallelism (within configurations). The implementation is described in full in Algorithm 2.

For the OpenML benchmark we used Algorithm 2 with $n_0 = 512$, $\eta = 4$ and $r_{min} = 1/4$. Since 3x3 nested cross-validation was used in this benchmark, the SH method was performed independently for each of the 3 outer folds. For each outer fold, we perform a cross-validated variant of Algorithm 2 in which Step 18 is performed across the 3 inner folds. Specifically, each configuration is trained and evaluated for every inner fold, and the validation loss used to rank the configurations is given as the mean across the 3 inner folds. For the Kaggle benchmark we used Algorithm 2 with $n_0 = 256$, $\eta = 4$ and $r_{min} = 1/16$. Additionally, we also leveraged the early-stopping functionality of the boosting frameworks, so that the training of each configuration may terminate early, if it is detected that the validation loss has not improved in the last 10 boosting iterations. Identical hyper-parameter ranges were used in both benchmarks, and are given in full in the next section.

# D  Hyper-Parameter Search Space

In Tables 3, 4, 5, 6 and 7 we list the hyper-parameter ranges for XGBoost, LightGBM, CatBoost, KTBoost and SnapBoost, respectively.

**Maximum depth.** LightGBM enforces a constraint on the maximum number of leaves that corresponds to a constraint on the maximum (complete) tree depth of 16. Furthermore, in our setup we had to further limit the maximum depth to 15 in order to avoid out-of-memory errors. For CatBoost, we had to limit the maximum depth to 16, for the same reason. The other three frameworks (XGBoost, SnapBoost and KTBoost) were able to use trees of depth up to 19 without any memory issues.

**Ordered boosting.** CatBoost offers both ordered boosting (`boosting_type=Ordered`) as well as standard boosting (`boosting_type=Plain`). All CatBoost experiments presented in the paper were performed using the default setting of ordered boosting. Since CatBoost seems to be slower than the other frameworks, afterwards we re-ran the experiments using `boosting_type=Plain` but were only able to see around a 20% improvement in runtime.

**KTBoost.** For KTBoost, we were unable to use the early stopping functionality as the library generated errors. In the main manuscript, we report the KTBoost results obtained for the Credit Card Fraud dataset. In the meantime, we additionally collected the KTBoost results for the Rossman Store Sales dataset: 54 hours (total tuning and evaluation time) vs. less than 30 minutes (SnapBoost), average test score (root mean squared error) 659.73 vs. 627.50 (SnapBoost).

Table 3: XGBoost hyper-parameter ranges.

| Hyper-parameter | Min | Max | Scale |
|---|---|---|---|
| max_depth | 1 | 19 | Linear |
| num_round | 10 | 1000 | Linear |
| learning_rate | -2.5 | -1 | Log10 |
| colsample_bytree | 0.5 | 1.0 | Linear |
| subsample | 0.5 | 1.0 | Linear |
| lambda | -2 | -2 | Log10 |
| tree_method | | hist | |
| max_bin | | 256 | |

Table 4: LightGBM hyper-parameter ranges.

| Hyper-parameter | Min | Max | Scale |
|---|---|---|---|
| max_depth | 1 | 15 | Linear |
| num_round | 10 | 1000 | Linear |
| learning_rate | -2.5 | -1 | Log10 |
| feature_fraction | 0.5 | 1.0 | Linear |
| bagging_fraction | 0.5 | 1.0 | Linear |
| lambda_l2 | -2 | -2 | Log10 |
| max_bin | | 256 | |

Table 5: CatBoost hyper-parameter ranges.

| Hyper-parameter | Min | Max | Scale |
|---|---|---|---|
| max_depth | 1 | 16 | Linear |
| n_estimators | 10 | 1000 | Linear |
| learning_rate | -2.5 | -1 | Log10 |
| subsample | 0.5 | 1.0 | Linear |
| l2_leaf_reg | -2 | -2 | Log10 |
| max_bin | | 256 | |
| boosting_type | | Ordered | |
| bootstrap_type | | MVS | |
| sampling_frequency | | PerTree | |
| grow_policy | | SymmetricTree | |

Table 6: KTBoost hyper-parameter ranges.

| Hyper-parameter | Min | Max | Scale |
|---|---|---|---|
| max_depth | 1 | 19 | Linear |
| n_estimators | 10 | 1000 | Linear |
| learning_rate | -2.5 | -1 | Log10 |
| subsample | 0.5 | 1.0 | Linear |
| max_features | 0.5 | 1.0 | Linear |
| theta | -1.5 | 1.5 | Log10 |
| alphaReg | -6 | 3 | Log10 |
| n_components | 1 | 100 | Linear |
| update_step | | newton | |
| nystroem | | True | |
| base_learner | | combined | |

Table 7: SnapBoost hyper-parameter ranges

| Hyper-parameter | Min | Max | Scale |
|---|---|---|---|
| num_round | 10 | 1000 | Linear |
| min_max_depth | 1 | 19 | Linear |
| max_max_depth | 1 | 19 | Linear |
| learning_rate | -2.5 | -1 | Log10 |
| subsample | 0.5 | 1.0 | Linear |
| colsample | 0.5 | 1.0 | Linear |
| lambda_l2 | -2 | -2 | Log10 |
| tree_probability | 0.9 | 1.0 | Linear |
| fit_intercept | 0 (False) | 1 (True) | Linear |
| alpha | -6 | -3 | Log10 |
| gamma | -3 | 3 | Log10 |
| n_components | 1 | 100 | Linear |
| hist_nbins | | 256 | |

# E    Tuned SnapBoost Hyper-parameter Values

In this section, the tuned hyper-parameter values (for the Kaggle benchmark) are provided for XGBoost, LightGBM, SnapBoost, CatBoost and KTBoost in Table 8, Table 9, Table 10, Table 11 and Table 12 respectively.

Table 8: Tuned hyper-parameter values for XGBoost

| Hyper-parameter | Credit Card Fraud | Rossmann Store Sales | Mercari Price Suggestion |
|---|---|---|---|
| max_depth | 1 | 16 | 19 |
| num_round | 303 | 923 | 804 |
| learning_rate | 0.037 | 0.018 | 0.035 |
| colsample_bytree | 0.568 | 0.552 | 0.679 |
| subsample | 0.866 | 0.906 | 0.548 |
| lambda | 0.283 | 2.124 | 6.102 |

Table 9: Tuned hyper-parameter values for LightGBM

| Hyper-parameter | Credit Card Fraud | Rossmann Store Sales | Mercari Price Suggestion |
|---|---|---|---|
| max_depth | 1 | 13 | 13 |
| num_round | 973 | 668 | 905 |
| learning_rate | 0.033 | 0.081 | 0.091 |
| feature_fraction | 0.631 | 0.631 | 0.846 |
| bagging_fraction | 0.857 | 0.965 | 0.680 |
| lambda_l2 | 0.057 | 0.014 | 6.719 |

Table 10: Tuned hyper-parameter values for SnapBoost.

| Hyper-parameter | Credit Card Fraud | Rossmann Store Sales | Mercari Price Suggestion |
|---|---|---|---|
| num_round | 593 | 878 | 800 |
| learning_rate | 0.027 | 0.046 | 0.045 |
| colsample | 0.644 | 0.693 | 0.845 |
| subsample | 0.650 | 0.667 | 0.764 |
| lambda_l2 | 1.558 | 33.409 | 2.001 |
| min_max_depth | 1 | 18 | 17 |
| max_max_depth | 1 | 19 | 17 |
| tree_probability | 0.912 | 0.925 | 0.934 |
| alpha | 0.004 | 0.011 | 0.016 |
| fit_intercept | false | true | true |
| gamma | 0.617 | 11.088 | 602.772 |
| n_components | 59 | 58 | 79 |

# F   NODE versus SnapBoost Benchmark

In this section, we compare SnapBoost with Neural Oblivious Decision Ensembles (NODE) [38]. NODE constructs deep networks of *soft* decision trees that can be trained using end-to-end back-propagation.

**Datasets.** For this benchmark we used 6 regression datasets as shown in Table 15. These datasets have approximately 10K examples and 20 features on average. We chose to use these relatively small datasets since, as we will see, training NODE is fairly slow. Firstly, we manually downloaded the data using the links provided in Table 15. For some datasets, we concatenated the provided train and test data files (*ailerons*, *elevators*, *puma32H*, and *bank8FM*). We used the concatenated matrices as input to the train/validation/test splitting during hyper-parameter optimization. As labels, we used column 41 for *ailerons*, column 20 for *parkinsons*, column 17 for *navalT*, column 19 for *elevators*, column 33 for *puma32h*, and column 9 for *bank8FM* (column indices being 1-based). We do not perform additional data preprocessing.

**Infrastructure.** The results in this section were obtained using a single-socket server with an 8-core Intel(R)Xeon(R) CPU E5-2630 v3 CPU, @2.40GHz, 2 threads per core, 64 GiB RAM, 2 NVIDIA GTX 1080 TI GPUs, running Ubuntu 16.04. We use NODE commit `3bae6a8a63f0205683270b6d566d9cfa659403e4` and PyTorch 1.4.0.

**Hyper-parameter optimization method.** To tune the hyper-parameters of NODE and SnapBoost, we used the optimization method described in Algorithm 2 with $n_0 = 1000$, $\eta = 4$ and $r_{min} = 1/20$. We tuned SnapBoost on the CPU using 16 single-threaded processes in parallel (`num_cores=16`).

Table 11: Tuned hyper-parameter values for CatBoost

| Hyper-parameter | Credit Card Fraud | Rossmann Store Sales |
|---|---|---|
| max_depth | 1 | 16 |
| n_estimators | 483 | 827 |
| learning_rate | 0.019 | 0.036 |
| subsample | 0.760 | 0.640 |
| l2_leaf_ref | 43.550 | 0.199 |

Table 12: Tuned hyper-parameter values for KTBoost

| Hyper-parameter | Credit Card Fraud |
|---|---|
| max_depth | 1 |
| n_estimators | 427 |
| learning_rate | 0.027 |
| subsample | 0.809 |
| max_features | 0.735 |
| theta | 0.125 |
| alphaReg | 0.139 |
| n_components | 56 |

NODE was tuned sequentially, one hyper-parameter configuration at a time, using both available GPUs. It was necessary to both GPUs since NODE crashed with out-of-memory errors when using only one.

In this benchmark, we used 2x2 nested cross-validation. The SH method was performed independently for each of the 2 outer folds. For each outer fold, we performed a cross-validated variant of Algorithm 2 in which Step 18 was performed across the 2 inner folds. Specifically, each configuration was trained and evaluated for every inner fold, and the validation loss used to rank the configurations was given as the mean across the 2 inner folds. The training and evaluation loss used in this benchmark was the root mean squared error (RMSE).

**Hyper-parameter search space.** Tables 13 and 14 show the hyper-parameter ranges used in this benchmark. NODE's layer dimension is computed as $\texttt{layer\_dim} = \lceil\texttt{total\_trees}/\texttt{num\_layers}\rceil$. Other NODE parameter settings: `nus=(0.7,1.0`, `betas=(0.95, 0.998)`, `optimizer=QHAdam`, `epochs=100`, and `batch_size=min(int(dataset.shape[0]/2), 512)`.

Table 13: SnapBoost hyper-parameter ranges.

| Hyper-parameter | Min | Max | Scale |
|---|---|---|---|
| num_round | 64 | 2048 | Linear |
| min_max_depth | 1 | 8 | Linear |
| max_max_depth | 1 | 8 | Linear |
| learning_rate | -3 | 0 | Log10 |
| subsample | 0.5 | 1.0 | Linear |
| colsample | 0.5 | 1.0 | Linear |
| lambda_l2 | -2 | -2 | Log10 |
| tree_probability | 0.9 | 1.0 | Linear |
| fit_intercept | 0 (False) | 1 (True) | Linear |
| alpha | -6 | 3 | Log10 |
| gamma | -3 | 3 | Log10 |
| n_components | 1 | 100 | Linear |
| hist_nbins | | 256 | |

Table 14: NODE hyper-parameter ranges.

| Hyper-parameter | Min | Max | Scale |
|---|---|---|---|
| num_layers | 1 | 8 | Linear |
| total_trees | 64 | 2048 | Linear |
| depth | 1 | 8 | Linear |
| tree_dim | 2 | 3 | Linear |

**Experimental results.** Table 15 shows the result of the benchmark. The table includes information about the datasets' characteristics, as well as the test RMSE (averaged over the 2 outer folds), and total experimental time for both NODE and SnapBoost. The time is reported in hours. SnapBoost achieves a lower test RMSE than NODE on 4 datasets, whereas NODE wins on the remaining 2 datasets. In terms of experimental time, SnapBoost is on average approximately 160 times faster than NODE.

Table 15: NODE vs. SnapBoost Benchmark.

| Name | Rows | Features | RMSE (Test) | | Time (hours) | | Speed-up |
|------|------|----------|-------------|----------|--------------|-----------|----------|
| | | | NODE | SnapBoost | NODE | SnapBoost | |
| ailerons [1] | 13750 | 40 | 0.000204 | **0.000157** | 38.25 | 0.34 | 112.5 |
| parkinsons[7] [9, 35] | 5875 | 19 | 0.001718 | **0.000868** | 22.30 | 0.15 | 150 |
| navalT [3, 15] | 11934 | 16 | 0.006941 | **0.000631** | 55.19 | 0.20 | 275.9 |
| elevators [6] | 16599 | 18 | 0.005099 | **0.002074** | 32.83 | 0.24 | 136.7 |
| bank8FM [2] | 8192 | 8 | **0.028717** | 0.031298 | 27.45 | 0.11 | 249.5 |
| puma32h [10] | 8192 | 32 | **0.006424** | 0.007629 | 23.70 | 0.39 | 60.7 |

# G  Reformulation of Algorithm 1 as Coordinate Descent

The definition of the $\mathcal{F}$ given in (2) dictates that any function $f \in \mathcal{F}$ can be expressed a weighted sum over functions belong to the base hypothesis class $\mathcal{H}$. Furthermore, by Assumption 3, every function in $\mathcal{H}$ can be expressed as a scalar multiplied by one of the functions belonging to finite set $\bar{\mathcal{H}}$. Thus, every $f \in \mathcal{F}$ has an equivalent representation as a weighted sum over the functions $b_j \in \bar{\mathcal{H}}$:

$$f(x_i) = \sum_{j=1}^{|\bar{\mathcal{H}}|} \beta_j b_j(x_i),$$

where $\beta \in \mathbb{R}^{|\bar{\mathcal{H}}|}$ and typically the vast majority of the coefficients $\beta_j$ are zero. Next, we introduce the matrix $B \in \mathbb{R}^{n \times |\bar{\mathcal{H}}|}$, with entries given by $B_{i,j} = b_j(x_i)$. Given this definition, a given function $f \in \mathcal{F}$ evaluated at $x_i$ can be expressed:

$$f(x_i) = \sum_{j=1}^{|\bar{\mathcal{H}}|} \beta_j B_{i,j} = B_i \beta,$$

where $B_i \in \mathbb{R}^{1 \times |\bar{\mathcal{H}}|}$ denotes the $i$-th row of $B$. Thus minimization (1) over domain (2) is equivalent to minimizing the following objective function over $\beta \in \mathbb{R}^{|\bar{\mathcal{H}}|}$:

$$L(\beta) = \sum_{i=1}^{n} l(y_i, B_i \beta)$$

The optimal coordinate to update at the $m$-th iteration, given randomly chosen subclass index $u_m$, is given by:

$$
\begin{aligned}
j_m &= \underset{j \in I(u_m)}{\arg\min} \left[ \min_{\sigma \in \mathbb{R}} L(\beta^{m-1} + \sigma e_j) \right] \\
&= \underset{j \in I(u_m)}{\arg\min} \left[ \min_{\sigma \in \mathbb{R}} \sum_{i=1}^{n} l(y_i, B_i \beta^{m-1} + \sigma B_{i,j}) \right] \\
&\approx \underset{j \in I(u_m)}{\arg\min} \left[ \min_{\sigma \in \mathbb{R}} \sum_{i=1}^{n} \left( l(y_i, B_i \beta^{m-1}) + g_i \sigma B_{i,j} + \frac{h_i}{2} \sigma^2 B_{i,j}^2 \right) \right] \\
&= \underset{j \in I(u_m)}{\arg\min} \left[ \min_{\sigma \in \mathbb{R}} \sum_{i=1}^{n} h_i \left( -\frac{g_i}{h_i} - \sigma B_{i,j} \right)^2 \right],
\end{aligned}
\tag{13}
$$

where the approximation is obtained by taking the second-order Taylor expansion of $l(y_i, B_i \beta^{m-1} + \sigma B_{i,j})$ around $l(y_i, B_i \beta^{m-1})$, with expansion coefficients given by $g_i = l'(y_i, B_i \beta^{m-1})$ and $h_i = l''(y_i, B_i \beta^{m-1})$. Note that this optimization problem is directly equivalent to (3) in the original formulation of Algorithm 1. For a fixed coordinate $j$, the inner minimization over $\sigma$ has a closed-form solution:

$$\sigma_j^* = -\frac{\sum_i g_i B_{i,j}}{\sum_i h_i B_{i,j}^2} = -\frac{\nabla_j L(\beta^{m-1})}{\nabla_j^2 L(\beta^{m-1})}, \tag{14}$$

where we have used two identities that link the first and second-order derivatives of $L(\beta)$ to the coefficients $g_i$ and $h_i$ as follows:

$$\nabla_j L(\beta^{m-1}) = \frac{\partial}{\partial \beta_j^{m-1}} \left( \sum_{i=1}^n l(y_i, B_i \beta^{m-1}) \right) = \sum_{i=1}^n g_i B_{i,j} \tag{15}$$

$$\nabla_j^2 L(\beta^{m-1}) = \frac{\partial^2}{\partial (\beta_j^{m-1})^2} \left( \sum_{i=1}^n l(y_i, B_i \beta^{m-1}) \right) = \sum_{i=1}^n h_i B_{i,j}^2. \tag{16}$$

Now, by plugging (14) into (13) we have:

$$
\begin{aligned}
j_m &= \arg\min_{j \in I(u_m)} \left[ \sum_{i=1}^n h_i \left( -\frac{g_i}{h_i} + \frac{\nabla_j L(\beta^{m-1})}{\nabla_j^2 L(\beta^{m-1})} B_{i,j} \right)^2 \right] \\
&= \arg\min_{j \in I(u_m)} \left[ -2 \frac{\nabla_j L(\beta^{m-1})}{\nabla_j^2 L(\beta^{m-1})} \sum_{i=1}^n g_i B_{i,j} + \left( \frac{\nabla_j L(\beta^{m-1})}{\nabla_j^2 L(\beta^{m-1})} \right)^2 \sum_{i=1}^n h_i B_{i,j}^2 \right] \\
&= \arg\min_{j \in I(u_m)} \left[ -\frac{(\nabla_j L(\beta^{m-1}))^2}{\nabla_j^2 L(\beta^{m-1})} \right] = \arg\max_{j \in I(u_m)} \left[ \left| \frac{\nabla_j L(\beta^{m-1})}{\sqrt{\nabla_j^2 L(\beta^{m-1})}} \right| \right],
\end{aligned}
$$

where in the third equality we have again used (15) and (16).

# H   Proof of Lemma 1

This proof is analogous to Proposition 4.3 in [36], adapted to use the norm induced by $\Phi$, as well as the second-derivative information. From the statement of the Lemma, we have the following definition of $\Gamma(\beta)$ for $j \in \left[ |\bar{\mathcal{H}}| \right]$:

$$\Gamma_j(\beta) = \frac{\nabla_j L(\beta)}{\sqrt{\nabla_j^2 L(\beta)}}$$

Now, given the definition of $j_m$ in (6), we have:

$$\mathbb{E}_m \left[ \Gamma_{j_m}(\beta^{m-1})^2 \right] = \sum_{k=1}^K \phi_k \max_{j \in I(k)} \Gamma_j(\beta^{m-1})^2 = \sum_{j=1}^{|\bar{\mathcal{H}}|} \lambda_j \Gamma_j(\beta^{m-1})^2,$$

where $\lambda_j$ is defined as follows:

$$
\lambda_j = \begin{cases}
\phi_1, & \text{if } j = \arg\max_{j \in I(1)} \Gamma_j(\beta^{m-1})^2 \\
\phi_2, & \text{if } j = \arg\max_{j \in I(2)} \Gamma_j(\beta^{m-1})^2 \\
\vdots & \\
\phi_K, & \text{if } j = \arg\max_{j \in I(K)} \Gamma_j(\beta^{m-1})^2 \\
0, & \text{otherwise.}
\end{cases}
$$

Now, by noting that $\sum_j \lambda_j = 1$ and $\lambda_j \geq 0$ and applying the Cauchy-Schwarz inequality:

$$
\begin{aligned}
\mathbb{E}_m \left[ \Gamma_{j_m}(\beta^{m-1})^2 \right] &= \left( \sum_j \lambda_j \right) \left( \sum_j \lambda_j \Gamma_j(\beta^{m-1})^2 \right) \\
&\geq \left( \sum_j \lambda_j |\Gamma_j(\beta^{m-1})| \right)^2 = ||\Gamma(\beta^{m-1})||_\Phi^2,
\end{aligned}
$$

where the last equality uses the definition of the $\Phi$-norm from Definition 1. $\qquad \square$

# I Proof of Theorem 2

From the update rule (4) and equations (5) and (6) we have:

$$L(\beta^m) = \sum_{i=1}^{n} l\left(y_i, B_i\beta^{m-1} - \epsilon\left(\frac{\nabla_{j_m}L(\beta^{m-1})}{\nabla_{j_m}^2 L(\beta^{m-1})}\right)B_{i,j_m}\right)$$

$$\leq \sum_{i=1}^{n} l(y_i, B_i\beta^{m-1}) - \epsilon\left(\frac{\nabla_{j_m}L(\beta^{m-1})}{\nabla_{j_m}^2 L(\beta^{m-1})}\right)B_{i,j_m}g_i$$

$$+ \frac{\epsilon^2}{2}\left(\frac{\nabla_{j_m}L(\beta^{m-1})}{\nabla_{j_m}^2 L(\beta^{m-1})}\right)^2 B_{i,j_m}^2 l''(y_i, z_i), \tag{17}$$

where the existence of the sequence $z_i$ are guaranteed by the Mean Value Theorem. Now, applying Assumption 1 and Assumption 2 we have for all $i \in [n]$:

$$\frac{l''(y_i, z_i)}{l''(y_i, B_i\beta^{m-1})} \leq \frac{S}{\mu} \implies l''(y_i, z_i) \leq \frac{S}{\mu}h_i, \tag{18}$$

where we recall that $l''(y, B_i\beta^{m-1}) = h_i$. Plugging into (17) we have:

$$L(\beta^m) \leq L(\beta^{m-1}) - \epsilon\left(\frac{\nabla_{j_m}L(\beta^{m-1})}{\nabla_{j_m}^2 L(\beta^{m-1})}\right)\sum_{i=1}^{n}B_{i,j_m}g_i + \frac{\epsilon^2}{2}\frac{S}{\mu}\left(\frac{\nabla_{j_m}L(\beta^{m-1})}{\nabla_{j_m}^2 L(\beta^{m-1})}\right)^2\sum_{i=1}^{n}B_{i,j_m}^2 h_i$$

$$= L(\beta^{m-1}) - \Gamma_{j_m}(\beta^{m-1})^2\left(\epsilon - \frac{\epsilon^2}{2}\frac{S}{\mu}\right) = L(\beta^{m-1}) - \frac{\mu}{2S}\Gamma_{j_m}(\beta^{m-1})^2, \tag{19}$$

where $\Gamma_j(\beta)$ is defined as in Lemma 1 and in the final step we have set the learning rate to be $\epsilon = \frac{\mu}{S}$.

Now we take the expectation of both sides of (19) with respect to the $m$-th iteration to attain:

$$\mathbb{E}_m\left[L(\beta^m)\right] \leq L(\beta^{m-1}) - \frac{\mu}{2S}\mathbb{E}_m\left[\Gamma_{j_m}(\beta^{m-1})^2\right]$$

$$\leq L(\beta^{m-1}) - \frac{\mu}{2S}\left\|\Gamma(\beta^{m-1})\right\|_\Phi^2$$

$$= L(\beta^{m-1}) - \frac{\mu}{2S}\left(\sum_{k=1}^{K}\phi_k \max_{j\in I(k)}\left|\frac{\nabla_j L(\beta^{m-1})}{\sqrt{\sum_{i=1}^{n}h_i B_{i,j}^2}}\right|\right)^2$$

$$\leq L(\beta^{m-1}) - \frac{\mu}{2S^2}\left(\sum_{k=1}^{K}\phi_k \max_{j\in I(k)}\left|\frac{\nabla_j L(\beta^{m-1})}{\sqrt{\sum_{i=1}^{n}B_{i,j}^2}}\right|\right)^2$$

$$= L(\beta^{m-1}) - \frac{\mu}{2S^2}\left(\sum_{k=1}^{K}\phi_k \max_{j\in I(k)}\left|\nabla_j L(\beta^{m-1})\right|\right)^2$$

$$= L(\beta^{m-1}) - \frac{\mu}{2S^2}\|\nabla L(\beta^{m-1})\|_\Phi^2 \tag{20}$$

where the second inequality follows from Lemma 1, the third inequality follows from Assumption 2, and the penultimate equality follows due to Assumption 3.

We then apply directly apply Proposition 4.4 and 4.5 from [36] (which in turn rely on Assumption 1) to obtain the following lower bound:

$$\left\|\nabla L(\beta^{m-1})\right\|_\Phi^2 \geq 2\mu\Theta^2\left(L(\beta^{m-1}) - L(\beta^*)\right), \tag{21}$$

where $\beta^*$ is the vector that minimizes $L(\beta)$. Now subtracting $L(\beta^*)$ from both sides of (20) and applying (21) we have:

$$\mathbb{E}_m\left[L(\beta^m) - L(\beta^*)\right] \leq L(\beta^{m-1}) - L(\beta^*) - \frac{\mu}{2S^2}\|\nabla L(\beta^{m-1})\|_\Phi^2$$

$$\leq \left(L(\beta^{m-1}) - L(\beta^*)\right)\frac{\mu^2}{S^2}\Theta^2$$

The proof is furnished by following a telescopic argument. □

## Footnotes

[5]https://www.kaggle.com/cast42/xgboost-in-python-with-rmspe-v2

[6]https://www.kaggle.com/tsaustin/mercari-price-recommendation

[7] https://archive.ics.uci.edu/ml/datasets/Parkinsons+Telemonitoring