[Reviews · NeurIPS 2020]

Review 1

Summary and Contributions: This paper proposes a boosting approach that - implements a Newton step at each iteration, - randomly chooses, at each iteration, a base learner among different classes, - is provably convergent - is specialized to the case when the base classifiers are decision trees of given depths and linear regressors, with a provided C++ implementation, called MixBoost. == Post-feedback Feedback taken into account, I stick to my score. Please be sure to take into account the remark on the 256 bins (i.e. state beforehand, at some point, that you work with a 8-bit representation).

Strengths: Strengths - an algorithmic strategy that allows the selection of base classifiers among different (structured) classes - a convergence proof of the algorithm, which subsumes existing results (those given in [31]) - numerical simulations

Weaknesses: Weaknesses: - the paper could gain of having a thorough discussion on the generalization properties of the proposed learning procedure (and if no gain, that should be stated as well); - is is not clear how (not) stringent is Assumption 3 regarding the hypothesis classes, whereas it is key for the results provided; in addition the finite class claim related to floating point arithmetic could be developed a bit more; - it is not clear to me what the main strength of MixBoost is: -- is it related to the computational savings? -- is it related to the generalization performances of the method? -- is it related to the use of Random Fourier Features? - regarding Random Fourier Feature: wouldn't an even accelerated method such as Fastfood (Le et al. 2013) preferable to the sampling proposed here?

Correctness: The claims made are correct, as far as I have checked them. Empirical methodology: it is correct, with the authors taking time to talk about statistical significance. There is a weird sentence: "The number of histogram bins h can be at most 256", where it is not clear why 256 should be the maximum number of bins.

Clarity: It is rather clearly written. Maybe, there is a lack of structure regarding the main ideas that have to be retained (cf. Mixboost and my remarks above).

Relation to Prior Work: There is a section devoted to the related work, section 1.1. (which could be unnumbered since there is no section 1.2).

Reproducibility: Yes

Additional Feedback: Dealing with Assumption 3 seems to be crux: the authors should explain how from a predefined family of base learners, a new class of models can be built that fulfill the requirement of Assumption 3. Typos: - line 117: b(x) -> b - line 223: samples from the Fourier transform of the Gaussian kernel -> random Gaussian vectors ?


Review 2

Summary and Contributions: In this paper the authors propose a general boosting method which relies on a multitude of learner classes in order to build the final classifier. The main method, coined HNBM, selects a single class at each iteration step and chooses the weak learner from that class. A practical ML tool MixBoost is derived from HNBM by fixing the learner classes (decision trees + a single linear model). Empirical results are provided showing the effectiveness of both HNBM and MixBoost.

Strengths: Combining several learner classes has been a common technique in practical boosting and ensemble methods in general, since it ensures a better diversity among the base classifiers, hence better performance. While the empirical results shown in this paper are not surprising to any ensemble learning practitioner, the strength of this work resides in providing a full theoretical setting for understanding and analyzing heterogeneous base learners. To the best of my knowledge, HNBM is the first framework that provides a clear theoretical insight on heterogeneous learners which englobes several learning paradigms, from heterogeneous data/attributes, to multi-view/multi-source learning. This by itself makes this contribution of significant interest for all the ML community. In particular, HNBM opens up several research questions (different probability mass functions, theoretical aspects of diversity in ensemble learning, etc.). A more practical contribution is MixBoost, which provides a powerful tool for heterogeneous boosting comparable to state of the art toolkits.

Weaknesses: I have a few comments on this paper, even though it would be unfair to call them weaknesses. They are listed below in no particular order. - It's regrettable that the probability mass function is practically unexploited. In MixBoost it is set to a quasi-uniform distribution, which depends on only one single parameter. Intuitively, each learner class should be considered individually, even in the case of BDT of different depths. I think that considering various probability mass function would've added further depth to the experimental setting (unless I'm missing an obvious reason why the quasi-uniform distribution is well suited...). - Continuing from the previous point, it would have been interesting to have a discussion on how the choice of the probability mass function influences the theoretical guarantees of in section 2.4. - The main strength of HNBM resides in using arbitrary mass functions, yet MixBoost only relies in BDT and LR. I strongly think that combining other types of classifiers should provide further insight on MixBoost.

Correctness: As far as I could tell, the theoretical results and analysis provided in this paper are sound. The empirical results in the main paper could've used a bit more breathing space, but the authors do a nice job at providing ample implementation details in the supplementary material.

Clarity: Very well written paper, easy to follow and the main ideas are clearly outlined. Some parts, such as Sec 2.3 might not be easy to access for the non expert reader.

Relation to Prior Work: The authors provide a nice review of existing works and, to the best of my knowledge, they cite most of the key papers related to the current work.

Reproducibility: Yes

Additional Feedback: after rebuttal: I'm satisfied by the authors answers, and as such I'm keeping my original rating for this paper.


Review 3

Summary and Contributions: The paper introduces a boosting method in which the hypothesis class may vary along the iterations. More precisely, at each iteration a base hypothesis class is chosen randomly from a fixed set of hypothesis classes. The learning algorithm is a Newton descent in a functional space. Some theoretical guarantees are provided. The method is applied using decision trees of variable size and linear regressors with random Fourier features as hypothesis classes. Promising experimental results are given.

Strengths: The paper is well written and easy to read. The paper is a new contribution on heterogeneous boosting. The proposed method improves performance over previous works based on the idea of a random selection of the base hypothesis class. Theoretical guarantees are provided. I did not check the proofs in detail but the proofs seem sound. The method is implemented with decision trees and linear regressors and prominsing experimental results are provided.

Weaknesses: In my opinion, the paper should contain a more thorough discussion on the choice of the hypothesis classes and on the probability distribution. Theoretically, the bound in Theorem 2 is an expectation over the subclass selection. The authors should discuss this issue and provide elements for choosing the subclasses and the probability distribution. Also, the role of $\Theta$ could be discussed in more details. As a consequence, the choice of the subclasses and of the probability distribution in MixBoost seems completely ad hoc and is not justified. For the experimental part, if the hyperparameter ranges can be found in the Appendix, the optimal values are not given. The paper should provide the optimal values (mainly D_min, D_max and p_t), a study of the robustness against the choice of the hyperparameter values and, if possible, should provide hints to choose the hyperparameter values.

Correctness: To the best of my knowledge, the claims are correct and the empirical methodology is correct. But, as said before, additional comments would be welcome and a robustness experimental study should be given.

Clarity: The paper is well written.

Relation to Prior Work: To the best of my knowledge, the related work is clearly discussed and compared with.

Reproducibility: No

Additional Feedback: Thanks for the feedback. While the authors have done a good job, I remain convinced that the choice of the hypothesis classes and of the PMF could have been more thoroughly discussed both theoretically and experimentally. For instance, it is surprising that D_min=D_max for every dataset. What would be the effect to consider one hypothesis class with small trees (say D_max=5) and large trees (say D_min=6 and D_max =20) and linear regressors. My evaluation is not changed and the paper can be accepted at NeurIPS. ******** The paper could be improved with more thorough discussion on the choice of the hypothesis classes and of the probability distribution both theoretically and experimentally. The choices done in MixBoost should be justified. For the experimental study, the optimal hyperparameters should be given. An experimental study should be given with varying value of $p_t$. For instance, I wonder what are the results for $p_t$=1.

[Author Response · NeurIPS 2020]

We begin by sincerely thanking all reviewers for carefully reading our paper and providing useful feedback. Below, we
respond to all concerns raised in the initial reviews. If accepted, we will amend the manuscript accordingly.

**Generalization properties of heterogeneous boosted ensembles (R2).** In the paper, we provide extensive *empirical*
evidence that such ensembles can improve generalization vs. homogenous boosted decision trees. From the *theoretical*
perspective, margin-based generalization bounds for heterogeneous boosted ensembles have been previously derived in
the DeepBoost paper (Cortes et al. 2014). Specifically, the authors showed that the generalization error is bounded
from above by the empirical margin error plus a weighted sum over the Rademacher complexities of the different base
hypothesis subclasses. The weights correspond to how many times each subclass appears in the ensemble. The authors
argue that heterogeneous ensembles have nice generalization properties, since by selecting a highly complex subclass
fairly *infrequently*, one may be able to reduce the empirical margin error without blowing up the complexity term.

**What is the role of $\Theta$ (R4)?** To gain more intuition, let us consider an alternate but equivalent definition:

$$\Theta = \min_{c \in Range(B)} \mathbb{E}_{k \sim \Phi} \left[ \max_{j \in I_k} \left[ \cos(B_j, c) \right] \right]$$

The term inside the expectation, for a given subclass $k$, measures the closest fit between a hypothesis in the subclass
and the direction $c$ (larger is better). Thus, $\Theta$ measures the worst-case *expected* closest fit, where the expectation is
taken over the PMF $\Phi$. The result of Theorem 2 tells us that larger values of $\Theta$ will lead to faster convergence.

**Can we use our theory to determine the best PMF and/or subclasses (R3, R4)?** Theorem 2 provides a *guarantee*
that HNBM will converge in terms of training loss. While the choice of PMF and/or subclasses affect the convergence
rate, it is not necessarily a good idea to explicitly optimize them for fast convergence. To see this, let us imagine there
exists a perfectly *dense* subclass with index $k^*$ for which, for all $c \in Range(B)$, there exists a $j' \in I_{k^*}$ such that
$\cos(B_{j'}, c) = 1$. In this case, our theoretical results imply that convergence rate can be maximized by assigning all mass
in the PMF to subclass $k^*$. However, a subclass with this property is likely to be very complex, which is not ideal from
the perspective of generalization (see above). Thus, it is preferable to parameterize the PMF and use cross-validation.

**Which hypothesis classes satisfy Assumption 3 in practice (R2)?** The role of Assumption 3 is to separate the *scale*
of the hypothesis (i.e., $\sigma$) from the *structure* of the hypothesis (i.e., $b$). This makes the derivations simpler and easier to
follow. The only restriction it places on the hypothesis subclasses in practice is that they satisfy a scalability assumption:
if $b$ belong to the subclass, then $\sigma b$ must along belong to the subclass for any $\sigma \in \mathbb{R}$. This is certainly true for
decision tree regressors, linear regressors and most other regressors we can think of. The argument that the number of
(normalized) hypotheses is finite when represented using machine precision is a common argument found in machine
learning textbooks (see for example Section 2.3.1 of the textbook by Shalev-Shwartz and Ben-David).

**What is the main strength of MixBoost (R2)?** The main strength of MixBoost is that it can achieve better generaliza-
tion than boosting machines that use trees alone (e.g. XGBoost, LightGBM) without significantly increasing the tuning
time required, which is a problem suffered by other heterogeneous boosting machines such as KTBoost.

**What motivated the choice of base hypothesis subclasses for MixBoost (R3, R4)?** Our choices for the base
hypothesis subclasses are motivated by existing ideas from the literature: using decision trees of varying depth is
inspired by DeepBoost (Cortes et al. 2014) and using LRFs was inspired by KTBoost's use of kernels (Sigrist 2019).

**What motivated the choice of the PMF used by MixBoost (R3, R4)?** This PMF was motivated by a desire to keep
the number of hyper-parameters low (in our case, it depends only on $p_t$, $D_{min}$ and $D_{max}$) whilst allowing a high-degree
of heterogeneity. We intend to pursue other, more general, choices for the PMF in our future research.

**Optimal hyper-parameters values (R4).** The optimal values for the parameters $D_{min}$, $D_{max}$ and $p_t$ for the Kaggle
datasets are provided in the table below. We see that while the decision trees used are fairly homogeneous, between 7%
and 9% of the hypotheses in the resulting ensembles correspond to LRFs rather than trees.

| Dataset | $p_t$ | $D_{min}$ | $D_{max}$ |
|---|---|---|---|
| Credit Card Fraud | 0.912 | 1 | 1 |
| Rossmann Store Sales | 0.925 | 18 | 19 |
| Mercari Price Suggestion | 0.934 | 17 | 17 |

**Results for $p_t = 1$ (R4).** The results labelled MIX-T in Figures 1 and 2 correspond exactly to the case where $p_t = 1$,
and the results labelled MIX correspond to the case where $p_t$ is tuned.

**Why is the maximum number of histogram bins 256 (R2)?** We use a single byte for performance reasons.

**Could the Fastfood technique (Le et. al 2013) by used instead of random Fourier features (R2)?** We were not
aware of this work until now but it certainly looks like it could be used to improve the MixBoost implementation further.

[Meta-Review · NeurIPS 2020]

This paper describes a boosting framework where for each iteration the classifier is randomly selected from a given set of classifiers. The paper provides some theoretical guarantees, and an empirical evaluation with some good results. Judging the results, it is really the version that combines trees and a linear regressor over random Fourier features, that works best. I suppose this supports the original idea, but will also warrant a more detailed future investigation.